# CRISPR/Cas9-Mediated Targeting of BPV-1-Transformed Primary Equine Sarcoid Fibroblasts

**DOI:** 10.3390/v15091942

**Published:** 2023-09-17

**Authors:** Anne Monod, Christoph Koch, Christoph Jindra, Maarten Haspeslagh, Denise Howald, Christian Wenker, Vinzenz Gerber, Sven Rottenberg, Kerstin Hahn

**Affiliations:** 1Swiss Institute of Equine Medicine (ISME), Department of Clinical Veterinary Medicine, Vetsuisse Faculty, University of Bern, 3001 Bern, Switzerland; vetswissmilitary@gmail.com (A.M.);; 2Institute of Animal Pathology, Vetsuisse Faculty, University of Bern, 3001 Bern, Switzerlandsven.rottenberg@unibe.ch (S.R.); 3Research Group Oncology, University Equine Clinic, University of Veterinary Medicine, 1210 Vienna, Austria; christoph.jindra@vetmeduni.ac.at; 4Department of Large Animal Surgery, Anesthesiology and Orthopaedics, Faculty of Veterinary Medicine, Ghent University, 9820 Merelbeke, Belgium; 5Zoo Basel, Binningerstrasse 40, 4054 Basel, Switzerland

**Keywords:** equine sarcoid, bovine papillomavirus, genome integration, episomes, Vimentin, CRISPR/Cas9

## Abstract

Equine sarcoids (EqS) are fibroblast-derived skin tumors associated with bovine papillomavirus 1 and 2 (BPV-1 and -2). Based on Southern blotting, the BPV-1 genome was not found to be integrated in the host cell genome, suggesting that EqS pathogenesis does not result from insertional mutagenesis. Hence, CRISPR/Cas9 implies an interesting tool for selectively targeting BPV-1 episomes or genetically anchored suspected host factors. To address this in a proof-of-concept study, we confirmed the exclusive episomal persistence of BPV-1 in EqS using targeted locus amplification (TLA). To investigate the CRISPR/Cas9-mediated editing of BPV-1 episomes, primary equine fibroblast cultures were established and characterized. In the EqS fibroblast cultures, CRISPR-mediated targeting of the episomal E5 and E6 oncogenes as well as the BPV-1 long control region was successful and resulted in a pronounced reduction of the BPV-1 load. Moreover, the deletion of the equine Vimentin *(VIM)*, which is highly expressed in EqS, considerably decreased the number of BPV-1 episomes. Our results suggest CRISPR/Cas9-based gene targeting may serve as a tool to help further unravel the biology of EqS pathogenesis.

## 1. Introduction

Papillomaviruses are non-enveloped viruses and are considered strictly host-specific animal and human pathogens. They usually cause benign, hyperplastic, epithelial lesions referred to as papillomas or warts. In addition, they are associated with the development of malignant tumors, including carcinomas of the skin, the oropharynx, and the genital region [1]. Unlike other papillomaviruses, bovine papillomavirus types 1 and 2 (BPV-1 and -2), and potentially also BPV-13, cross the species barrier and cause fibroblast-derived skin tumors called EqS in horses, donkeys, mules, zebras, and other wild equids [2,3,4,5,6]. EqS are the most prevalent skin tumors in equids worldwide and are characterized by their potential for locally invasive and unpredictable growth, including high recurrence rates after surgical resection [7,8,9,10].

Based on Southern blotting analysis, the BPV-1 genome was not found to be integrated in the host cell genome; instead, it persists as episomes in the nucleus of transformed EqS tumor cells [11]. This finding, however, has not been confirmed by state-of-the-art sequencing technologies. BPV-1 episomes encompass 8 kb and are composed of a circular, double-stranded DNA genome comprising the long control region (LCR), that controls transcription and replication of the viral DNA as well as six early (E), regulatory or transforming, and two late structural genes. Among the early genes, E5 and E6 encode the major BPV oncoproteins that promote fibroblast hyperproliferation, induce invasive growth in vitro, and enhance cell anchorage-independent growth [12,13].

In contrast, in human papillomavirus (HPV)-associated neoplasia, integration of the virus into the host genome is considered a crucial step for uncontrolled oncoprotein expression and malignant progression [14,15,16]. Gene editing CRISPR/Cas9 technology has been used successfully to induce frameshift mutations in the genome-integrated HPV E6 and E7 oncogenes as well as the LCR in cervical carcinoma-derived SiHa, HeLa, or CaSki cells [17,18,19,20,21,22]. This reduced oncogene transcription, impeded cervical carcinoma cell proliferation, or resulted in senescence.

In our proof-of-concept study, we used a similar approach to target BPV-1 episomes in primary equine sarcoid fibroblast cultures. We initially applied TLA and confirmed the absence of BPV-1 and -2 genome integration in EqS using a sequencing-based approach. We further established EqS fibroblast cultures that were used to target BPV-1 episomes and the equine *VIM* encoding a host protein strongly expressed in EqS [23]. CRISPR/Cas9-mediated targeting of the episomal E5 and E6 oncogenes as well as the LCR resulted in a pronounced reduction of the BPV-1 load. Moreover, the deletion of the equine *VIM* decreased the number of BPV-1 episomes. Our results suggest that CRISPR/Cas9-based gene targeting might be a useful tool to help unravel the biology of EqS pathogenesis.

## 2. Materials and Methods

### 2.1. Sample Collection

Surgically excised sarcoid tumor tissue from two horses, collected at the ISME Equine Clinic Bern, and from three captive wild equids two Somali wild donkeys (*Equus africanus somaliensis*) and one onager (*Equus hemionus*), collected from European zoos [6], was used to investigate the BPV-1 and BPV-2 host genome integration. For EqS-derived primary cell cultures, tumor tissue of EqS was obtained immediately following surgical excision from five horses treated for EqS disease at the ISME Equine Clinic Bern, Switzerland. Three of the lesions were fibroblastic [24] (EqS#1, EqS#3, and EqS#5) and two lesions were nodular- type [24] EqS tumors (EqS#4 and EqS#6). Control fibroblasts (ctrl Fbs) were collected from dermal samples obtained from a horse without EqS immediately following euthanasia for reasons unrelated to EqS disease or this study. All tissue samples were taken with a signed informed owner consent and ethical approval was granted by the institutional and cantonal officials for the use of excised tumor tissues (BE110/15). Histopathology was performed for all EqS samples. Appendix A summarizes the clinical data of the horses included in the study.

### 2.2. Targeted Locus Amplification

Assessment of BPV integration into the host genome was performed by Cergentis B.V. (Utrecht, The Netherlands) using the TLA technology [25]. In brief, crude nuclear extracts of five cryopreserved sarcoid tumors, that were previously tested BPV-1 positive by conventional PCR, and their matched normal skin control were processed for TLA using BPV-1- and BPV-2-specific primer pairs (Set 1: BPV-E1-Fw:5′-TTGTTCCTTTGTAGCTTCCA-3; BPV-E1-Rv:5′-TTGCTACCTTTATCGTTTGC-3′; Set: 2 BPV-E7: Fw: 5′-CTCAGATTTGGACCTGTTGT-3′, BPV-E7 Rv: 5′-GGAAGTCTGAAATCGGGTG-3′). PCR products were purified (Qiagen, Hilden, Germany), amplicons were library-prepped using the Illumina NexteraXT protocol and sequenced using an Illumina Miniseq sequencer (Illumina, San Diego, CA, USA). The reads were mapped using the Smith–Waterman alignment tool BWA-SW [26]. This tool allows for partial mapping, and it is optimally suited for identifying break-spanning reads.

### 2.3. Cell Culture

Primary EqS and ctrl Fbs were established from tissue samples excised from the core of the tumor or the dermis, respectively. Each tissue sample was minced (1 mm^3^ pieces), washed in PBS, and frozen at −150 °C in 50% DMEM (high glucose, L-glutamine; Gibco/Thermo Scientific, Waltham, MA, USA), 40% FCS (Gibco/Thermo Scientific) and 10% DMSO (Merck Millipore, Darmstadt, Germany). For the setup of cell cultures, tissue samples were thawed, washed twice in DMEM, and digested at 37 °C for 60 to 80 min in DMEM containing collagenase IV (2 mg/mL, MilliporeSigma, Darmstadt, Germany). After digestion, the tissue was mechanically dissociated using four flame-constricted Pasteur pipettes with decreasing luminal diameter, filtered (100 µm cell strainer; Thermo Fisher Scientific, Waltham, MA, USA) and washed twice in DMEM with 10% FCS and 1% penicillin-streptomycin (Gibco/Thermo Scientific). Cells were cultured in T150 cell culture flasks (TPP, Trasadingen, Switzerland) at 37 °C and 5% CO_2_. For trypsinization, 0.25% trypsin (Gibco/Thermo Scientific) was used. All primary cultures were tested negative for mycoplasma using the PlasmoTestTM—Mycoplasma Detection (InvivoGen, Toulouse, France).

### 2.4. BPV-1 PCR and rtPCR

All primary EqS fibroblast cultures were assessed for the presence of BPV-1 E2, E5, E6, E7, and L1 genes prior to each experiment. Equid ABL1 PCR was performed to confirm PCR compatibility of the DNA or the cDNA. DNA or RNA was isolated from 1 Mio cells per culture or from EqS tissue using the QIAamp^®^ DNA Mini Kit (Qiagen, Germany) or RNeasy Mini Kit (Qiagen), respectively. Reverse transcription was performed using Oligo(dT)15 Primer (Promega, Dübendorf, Switzerland) and corresponding reagents. Specific PCR conditions and primers [4,27] are listed in Appendix A. All PCR reactions were performed with GoTaq G2 Green MasterMix (Promega). The PCR products (10 µL) were separated by 2% TAE gel electrophoresis. Gels were visualized with U:Genius (Syngene, Cambridge, UK).

### 2.5. Lentivirus Production

Lentivirus production was performed as described with minor modifications [28]. Briefly, target loci were analyzed in all EqS-derived primary fibroblast cultures by Sanger sequencing (Microsynth, Balgach, Switzerland) prior to the design of the single guide (sg) RNAs. Subsequently, a non-targeting (NT) DNA oligonucleotide and oligonucleotides targeting *VIM* and BPV-1 E6, E5, and the LCR were designed (Benchling online tool, Benchling, Zürich, Switzerland; Table 1) and synthesized (Microsynth, Switzerland). Oligonucleotides were then melted at 95 °C for 5 min and annealed at room temperature for 2 h. Quick-ligase (NEB, USA) was used for the insertion into the BsmBI-digested (Fermantas/ThermoScientific) pLentiCRISPRv2 (Addgene_52961, USA) backbone. Vectors were transformed in E. coli (Endura™ Chemically Competent, Lucigen, Middleton, WI, USA) and plasmid isolation was performed using the Zyppy Plasmid Miniprep Kit (Zymo Research, Irvine, CA, USA). Cloning success was confirmed by sequencing (Microsynth). Clones with the correct insert were amplified and plasmids for lentivirus production were purified using the ZymoPURE II Plasmid Midiprep Kit (Zymo Research, USA). Lentivirus production was performed using calcium-phosphate-mediated transfection of HEK-293 T-cells (15 cm dish, Corning, Corning, NY, USA) with the plentiCRISPRv2 vector containing the respective sgRNA or a non-targeting sgRNA and the lentiviral plasmids pCMV-VSV-G, pMDLg/pRRE and the REV plasmid (Addgene, Watertown, MA, USA). The culture medium was changed 16 h after transfection. Lentivirus containing cell culture supernatant was harvested 48 and 64 h after transfection and concentrated via ultracentrifugation (22,000 RPM, 2 h, 4 °C, Rotor AH-629, Ultracentrifuge CP100NX, HITACHI, Tokyo, Japan). Virus was stored at −80 °C. Titers were determined using the ABM qPCR Lentivirus Titration Kit (Applied Biological Materials, Richmond, BC, Canada) according to manufacturer’s instructions. Lentiviral titers ranged from 10^8^ to 10^9^.

### 2.6. Transduction of EqS and Control Fibroblasts

For transduction of EqS and ctrl Fbs, cells in passage 5 were used for CRISPR-mediated targeting of BPV-1 oncogenes and passage 7 for *VIM* knockout experiments, respectively. EqS and ctrl Fbs were seeded in 6-well plates (TPP) and transduced 24 h post seeding at a cell confluence of 70% with a multiplicity of infection (MOI) of 50. Lentivirus encoding non-targeting (NT) sgRNA was used as a control. Polybrene (Merck Millipore, Germany) was added at a concentration of 8 µg/mL. Medium was changed after 30 h, and transduced cells as well as one well of non-transduced control cells were puromycin (5 µg/mL, Gibco/Thermo Scientific, USA) selected for 5 days. Cultures were trypsinized once and seeded in a T25 flask (TPP). Cells for DNA isolation (0.5–1 Mio) were collected when confluence was achieved or after growth arrest for 6 days. The target site modifications were assessed by Tracking of Indels by Decomposition (TIDE) analysis.

### 2.7. TIDE-PCR and TIDE Analysis

TIDE analysis was performed to analyze the CRISPR/Cas9-mediated spectrum of insertions and deletions [29]. DNA was isolated from cell pellets as described for the BPV-1 PCR. Target loci were PCR amplified with the Phusion High-Fidelity PCR Master Mix (Thermo Scientific) with HF Buffer. The primers used for TIDE analysis and PCR conditions are listed in Appendix A. The QIAquick PCR purification kit (Qiagen, Germany) was applied for the purification of PCR products. Sanger Sequencing (Microsynth) was performed using the forward TIDE primers. Target modifications were analyzed using the TIDE algorithms. Alignments were performed using Clustal Omega [30].

### 2.8. Proliferation Assays

Proliferation assays were performed as described using Cell Titer-Blue (CTB; Promega) according to the manufacturer’s instructions [28]. Briefly, 3000 cells in 200 µL medium were seeded per well in triplicate on 96-well plates (TPP) at day 0. Medium without cells was plated as a control. For each measurement time point, one plate was seeded. At intervals of 24 h, 20 µL CTB was added per well and incubated for 4 h. Measurements were performed at 560 Ex/590 Em nm on an Enspire Multimode Plate Reader (PerkinElmer, Singapore) and normalized to solvent control. All proliferation assays were performed in three technical and two biological replicates.

### 2.9. Immunofluorescence

To investigate *VIM* targeting on protein levels, immunofluorescence was performed according to established protocols [31] using 96-well plates and 12,000 cells per well. The Anti-Vimentin V9 antibody (1:1500, Abcam, Cambridge, UK) was used as the primary, and Alexa Fluor 488 IgG H+L as a secondary antibody (1:2000, Invitrogen/Thermo Fisher Scientific, USA). Three wells were incubated with the secondary antibody only as a control for unspecific binding. DAPI (Thermo Fisher Scientific, 1:60,000) was applied for nuclear staining. Fluorescent images were taken using a Delta Vision wide-field microscope (Cytiva/GE Healthcare Life Sciences, Chicago, IL, USA). The percentage of Vimentin-positive cells was evaluated in three high-power fields (40x, min 130 cells) per well (Fiji Cell Counter) [32]. IF was performed in three technical and two biological replicates.

### 2.10. Quantitative Real-Time PCR

To quantify the amount of BPV-1 DNA per cell and to compare two previously described quantitative real-time polymerase chain (qPCR) methods, samples were analyzed at the University of Veterinary Medicine, Vienna, Austria [33,34,35] and the University of Veterinary Medicine, Ghent, Belgium [36]. A BPV-1 E2 (Vienna) or BPV-1 E1 (Ghent), as well as luteinizing hormone subunit beta (*LHB*) or interferon beta-specific probes as reference genes were used to determine the viral load per cell. Quantified dilution series and negative controls were included in all runs. All samples were analyzed in duplicate on the same plate. For analysis, the mean of the two measurements was used. For the comparison of the viral load in the tumor tissue and cell cultures, cells were expanded directly after the generation of the primary cell line. For the *VIM* and BPV targeting experiments, cryopreserved cells were used.

### 2.11. Quantification and Statistical Analysis

Statistical analysis and graphical visualization were performed using GraphPad prism 8.4.2 software (GraphPad Software, Boston, MA, USA). For the *VIM* targeting, immunofluorescence and TIDE data were analyzed using ANOVA with Tukey’s multiple comparisons test.

## 3. Results

### 3.1. Targeted Locus Amplification Does Not Detect BPV-Integrations into the Host Cell Genome

Before testing CRISPR/Cas9-mediated genome editing of BPV-1-transformed equine tumor cells, we aimed at a sequencing-based confirmation of exclusively episomal BPV-1 and BPV-2 persistence. We used TLA with BPV-1- and BPV-2-specific primers, allowing the amplification and next generation sequencing of viral sequences and their neighboring integration sites [37]. In a pilot experiment, we screened two horse and three donkey sarcoids, as well as skin from a healthy horse using BPV-1- and BPV-2-specific primers. All sarcoids were tested BPV-positive by this assay and were used for TLA. Subsequent mapping identified amplicons as corresponding to BPV-1/BPV-2 sequences (Appendix A). A search for break-spanning reads that would indicate a genomic integration site yielded no results. The reads on the outer ends of the BPV sequence revealed an episomal organization. In line with this finding, mapping of the obtained sequences to the equine genome did not produce any results (Appendix A), confirming episomal BPV persistence in EqS.

### 3.2. BPV-1 Viral Load, Long Term Passaging, and Proliferation of Primary EqS Fibroblasts

To investigate the potential of CRISPR/Cas9-mediated targeting of BPV-1 episomes, we generated five primary EqS-derived fibroblast cultures and a control cell culture from BPV-1 free dermal equine fibroblasts. Ctrl Fbs and EqS cells displayed a mesenchymal, fibroblastic morphology indicating purity of the cell cultures (Appendix A). Tissue samples and fibroblasts derived thereof were subsequently analyzed by conventional, reverse transcription and qPCR for the presence and relative amount of selected BPV-1 genes.

BPV-1 E2, E5, E6, E7, and L1 DNA and RNA were detected in all tumor tissue samples and corresponding sarcoid fibroblast cultures. Non-infected fibroblasts scored negative (Appendix A). BPV-1 qPCR for E2 and E1 yielded divergent results. The E2-specific reaction detected higher viral loads in all samples compared to the E1-based approach. For the EqS tumor tissue, the E2-based qPCR revealed the highest BPV-1 load in EqS#6 followed by EqS#4, EqS#1, EqS#3, and EqS#5. The E1-based qPCR detected the highest BPV-1 loads in EqS#1, EqS#5, EqS#6, EqS#4, and EqS#3. Due to the divergent results, both qPCR methods were used in all subsequent experiments.

All EqS fibroblast cultures had markedly lower BPV-1 loads compared to the EqS tumor tissue. Furthermore, cell passaging led to a gradual decrease (EqS#1, 3, 5, 6) or loss (EqS#4) of BPV-1 E2 and E1 DNA until passage 6. At passage 12, BPV-1 was not detected in EqS#1, 5, and 6 anymore (Figure 1A). Hence, subsequent CRISPR/Cas9-based experiments were performed with EqS#1 and EqS#5 from passage 5 containing sufficient viral DNA. In long-term culturing, EqS#1 and EqS#5 were passaged over 45 times despite the successive loss of BPV-1 DNA. EqS#3 and EqS#4 and ctrl Fbs displayed a growth arrest in passage 5, 14, and 32, respectively.

When comparing primary cell proliferation rates, EqS#4 derived from a nodular sarcoid displayed the highest, and EqS#5 originating from a fibroblastic sarcoid displayed the lowest proliferation (Figure 1B).

### 3.3. CRISPR-Mediated Targeting of BPV-1 E5, E6, and LCR Reduces BPV-1 Loads in EqS Fibroblasts

To investigate the CRISPR/Cas9-mediated targeting of BPV-1 episomes, we designed sgRNAs for E5-, E6-, and the LCR and confirmed the sequence of the target locus in EqS tumor tissue (Appendix A). We then performed a transduction of EqS#1, 4, and 5 fibroblasts with lentiviral GFP vectors. EqS#4 did not survive the transduction, whereas EqS#1 and EqS#5 were efficiently transduced. We subsequently used EqS#1 and EqS#5 in passage 5 for the transduction with vectors encoding Cas9 and NT- or E5-, E6-, and LCR-specific sgRNAs. EqS#1-sgE5 fibroblasts reproducibly died 3 days after the transduction. Other sgRNA-transduced cells were viable but stagnated in growth thus confining further expansion. EqS fibroblasts transduced with BPV-1 targeting sgRNAs displayed a decrease of viral DNA levels of ≥50% compared to NT-sgRNA cells (Figure 2A), indicating a degradation of episomes induced by CRISPR/Cas9-mediated double-strand breaks.

To investigate the induction of frameshift mutations in BPV-1 episomes, TIDE analysis was conducted. TIDE analysis identifies the major induced mutations in the projected editing site and accurately determines their frequency in a cell population [29]. TIDE analysis revealed frameshift mutations in 80% of BPV-1 episomes on average in EqS#5-sgE6- transduced cells (Figure 2B). In EqS#5-sgE5 and EqS#6-sgLCR cells, frameshift mutations were present in 34% and 12% of BPV-1 episomes. In EqS#1, a low percentage of BPV-1 episomes harbored frameshift mutations (EqS#1-sgE6: 10%; EqS#1-sgLCR: 2%). Assessment of sgRNA target sequences did not indicate point mutations, impairing sgRNA binding (Appendix A). The contingency tables for the individual replicates of the experiment are shown in Appendix A.

### 3.4. CRISPR-Mediated Targeting of Vimentin Reduces BPV-1 Loads in EqS Fibroblasts

A classical hallmark of normal and transformed fibroblasts is the expression of VIM, a type III intermediate filament protein and major cytoskeletal component. Although VIM has frequently been used as a marker for mesenchymal neoplasms, its pathophysiological role in EqS fibroblasts is unclear. In nearly all human cell lines tested, *VIM* is not essential for cellular growth [38]. Therefore, the effects of a CRISPR/Cas9-mediated *VIM* targeting on BPV-1 loads were investigated in EqS fibroblasts.

We defined an sgRNA that reliably and specifically targets the *VIM* locus in EqS cell cultures. Using a *VIM* exon 1-specific sgRNA (sgVIM) TIDE analysis revealed highly efficient induction of frameshift mutations in ctrl Fbs and the EqS#1 and EqS#5 fibroblasts (Figure 3A). Appendix A shows the contingency tables for all replicates. Immunofluorescence confirmed efficient *VIM* targeting and revealed a significant reduction (*p* < 0.0001) of strongly VIM expressing cells in the ctrl Fbs and EqS#1 cells (Figure 3B–D).

To investigate the maintenance of the VIM targeting over long-term culturing, we performed a second TIDE analysis in a higher passage of the VIM-sgRNA-transduced ctrl Fbs and EqS#1 and EqS#5 cells. In ctrl Fbs-sgVIM, a strong selection in favor of *VIM* wild- type cells was revealed, whereas in VIM-sgRNA-transduced EqS fibroblasts, this effect was not detectable (Figure 3E).

QPCR showed a five times lower BPV-1 load in EqS#5-sgVIM fibroblasts compared to EqS#5-NT-sgRNA. In EqS#1-sgVIM, BPV-1 was not detectable (Figure 3F). In both VIM-sgRNA-transduced EqS cultures, the *VIM* targeting considerably impaired cell proliferation. EqS#5-sgVIM fibroblasts displayed a growth arrest confining further expansion. EqS#1-sgVIM proliferated slower compared to the corresponding NT-sgRNA-transduced cells (Figure 3G).

## 4. Discussion

In this study, we investigated BPV integration in the host genome in EqS and assessed if CRISPR/Cas9 is capable of targeting BPV-1 episomes as well as the equine VIM gene in primary EqS fibroblast cultures. TLA sequencing did not reveal complete BPV-1 or -2 DNA integrations into the EqS genome. These results are in line with the previously described lack of genome-integrated BPV in EqS [11]. The data further support that we exclusively targeted episomal BPV-1 in our LCR, E5, and E6 CRISPR/Cas9 editing experiments in primary EqS-derived fibroblast cultures.

To characterize our primary EqS cultures, we determined the BPV-1 load in the sarcoid tumor tissue and the corresponding EqS cells. This assessment revealed pronounced differences between the E1- and E2-specific qPCRs conducted in two different laboratories. This may be due to technical reasons. Alternatively, structural alterations of BPV-1 episomes such as duplications or deletions might have contributed to discrepancies of qPCR results. In line with this, dimer/multimer or hybrid episomes have been detected in three-quarters of HPV-16-associated head and neck squamous cell carcinomas [39]. Moreover, BPV-1 episomes with partially deleted sequences and intratypic sequence variation were found in EqS tissue [11,40,41]. The method-related differences need to be considered when comparing BPV-1 DNA levels in EqS between different studies.

Independent of the method, our analysis revealed pronouncedly decreased BPV-1 DNA loads in EqS fibroblast cultures compared to the tumor tissue. When we assessed the association of BPV-1 loads and EqS fibroblast proliferation, we did not observe that EqS cultures with high BPV-1 have growth advantages compared to cells with lower episome contents. Moreover, the BPV-1 negative ctrl Fbs derived from dermal equine fibroblasts proliferated faster than EqS cells. This is in contrast to previous reports using similar culture conditions [12]. These unexpected results might be related to the heterogeneous distribution of BPV-1 genomes in EqS tissue. In addition, a gradual loss of BPV-1 episomes upon cell proliferation needs to be considered. In line with this, a highly sensitive in situ hybridization (ISH) assay using a combined E5-, E6-, and E7-RNA scope probe detected BPV-1 mRNA/DNA in maximally 50% of transformed EqS fibroblasts of equine sarcoid tumors. Using a less sensitive ISH protocol, the signal was accentuated only at the dermo–epidermal junction [42,43]. Hence, the EqS culture preparations might contain different ratios of non-infected EqS fibroblasts, which might proliferate slower than our control dermal fibroblasts in vitro. However, in our study, ctrl Fbs displayed a growth arrest at passage 32 that was most likely associated with senescence. In contrast, EqS#1 and EqS#5 were expanded over 45 passages, despite the lack of E1 and E2, and thus probably the entire viral episome. The successful long-term passaging despite the absence of BPV-1 episomes might reflect that epigenetic changes and/or additional mutations occur in EqS fibroblasts that drive tumor growth in a BPV-1 independent manner.

The gradual loss of BPV-1 episomes over passages needs to be considered in experiments using primary EqS cultures and comprised a challenge when we performed our proof-of-concept study and assessed the CRISPR/Cas-based targeting of BPV-1 episomes in primary EqS cell cultures.

The in vitro targeting of E5, E6, and the LCR revealed that all sgRNAs used were efficient in reducing but not eliminating BPV-1 episomes.

Hence, CRIPSR/Cas9 can not only be used to target integrated papillomavirus sequences, as shown previously for HPV [17,18,19,20,21,22], but also papilloma virus episomes. Therefore, CRISPR/Cas9-based in vitro approaches represent a valuable tool to study the pathogenesis of papillomavirus-associated diseases, complementing the earlier described approaches using siRNA [44].

Interestingly, TIDE analysis revealed frameshift mutations in BPV-1 episomes at the CRISPR/Cas9 target locus. This suggests a repair of BPV-1 episomes by the error-prone DNA repair pathway of non-homologous end joining, as previously observed for integrated HPV [18]. Our data show that mainly frameshift and also in-frame mutations occur more frequently in CRISPR/Cas9-targeted fibroblast cultures containing high levels of BPV-1 episomes. Therefore, CRISPR/Cas9 editing may not only result in mutations that lead to the desired gene inactivation but also carry the risk of promoting the development of modified viruses in situations where the virus completes a full productive life cycle. This is an important aspect when discussing gene-editing approaches for the treatment of papillomavirus-associated diseases. Moreover, this could be exploited in an experimental setting to further elucidate the controversially debated question if BPV actually completes a productive life cycle in sarcoid-affected equids.

When using CRISPR/Cas9 to target the equine *VIM* gene, we achieved high knockout efficacies, as indicated by a high percentage of frameshift mutations in the TIDE analysis. VIM is considered a key intermediary filament of fibroblasts that interacts with the actin cytoskeleton and modulates the invasion of mesenchymal tumors [45,46,47,48,49]. As expected, VIM-gRNA-transduced EqS and ctrl Fbs displayed reduced proliferation or growth stagnation. The antiproliferative effect on ctrl Fbs rather excludes VIM as a therapeutic target for EqS. In long-term cultures, we observed a strong selection in favor of *VIM* wt cells in ctrl Fbs. This selection was absent in EqS cells suggesting a higher dependence of wt fibroblasts on VIM. VIM is a component of focal adhesions (FA) that mediate the cell attachment to the extracellular matrix [48]. Hence, these results might indicate a higher dependence of wild-type fibroblasts on FA integrity. This is in line with previous studies describing the anchorage-independent growth of EqS-derived cells in vitro compared to control fibroblasts [12]. In addition, we found that targeting *VIM* was associated with a decrease in BPV-1 DNA loads. In vitro studies revealed an association of VIM with circular, extrachromosomal DNA in the nucleus [50]. Accordingly, our results suggest a potential relevance of VIM for BPV-1 episomal maintenance. Hence, CRISPR/Cas9-based modulation of non-essential host genes, such as *VIM*, can be an additional, valuable method to gain new insights into PV-associated disease pathogenesis. In this regard, our proof-of- concept study might be expanded to a CRISPR screening approach using gRNA libraries [51]. This enables to functionally test which host genes are essential for BPV infection and may thereby provide new mechanistic insights into the disease.

## 5. Conclusions

We confirmed the long-standing dogma that BPV-1 DNA does not integrate into the host cell genome in EqS. Moreover, the results of our study underlined that a thorough characterization of primary EqS cells and assessment of BPV-1 loads over passages is essential. We further demonstrated that BPV-1 episomes and host genes can efficiently be targeted by using CRISPR/Cas9 in primary EqS fibroblast cultures. With this proof-of-concept study, we established an in vitro system that can be used to address open questions in EqS pathogenesis.

## Figures and Tables

**Figure 1 viruses-15-01942-f001:**
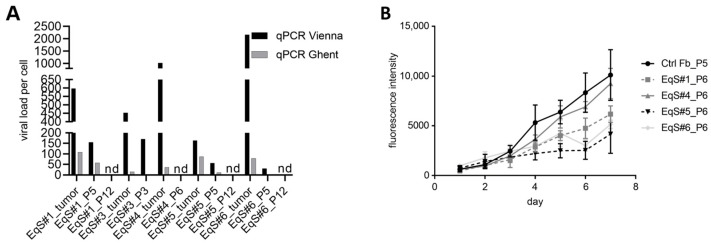
BPV-1 load in EqS tissue, corresponding cell cultures, and proliferation of EqS-derived fibroblasts. (**A**) The BPV-1 load in the EqS#1 and EqS#5 cell lines was determined by two different qPCR methods (G: Ghent, E1 and V: Vienna, E2). The BPV-1 load was higher for the tumor tissue compared to the corresponding EqS fibroblast cultures. All EqS fibroblast cultures displayed a gradual loss of BPV-1 DNA over passages. (**B**) EqS#4 showed the highest proliferation followed by EqS#1, 6, and 5. Note that the ctrl Fbs proliferated faster than the EqS-derived cells. The figure shows the means and standard deviations from three technical replicates and two biological replicates. Ctrl Fbs: BPV-1 negative control fibroblasts. EqS#1, 4, 5, 6: equine sarcoid-derived cells. nd: not detectable or lower than 0.001.

**Figure 2 viruses-15-01942-f002:**
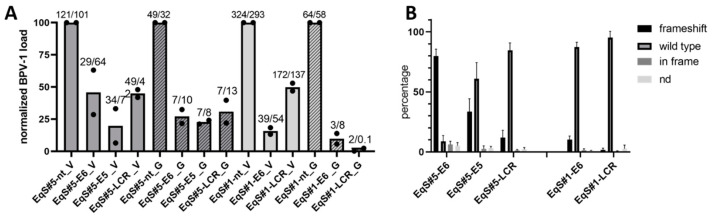
CRISPR/Cas9-mediated targeting of BPV-1 E6, E5, and the LCR in EqS fibroblasts. (**A**) Quantification of the BPV-1 load by two different qPCR methods (V = Vienna, E2; G = Ghent, E1) revealed a pronounced reduction of the BPV-1 DNA load in EqS#5-sgE5 as well as EqS#5 and EqS#1 transduced with E6- and LCR-targeting sgRNA. Note that EqS#1-sgE5 cells were not viable. The plot shows means and values (dots) from two biological replicates normalized to the viral load in NT-sgRNA-transduced cells. The numbers above the columns display absolute BPV-1 DNA loads per cell in each replicate. (**B**) TIDE analysis revealed high numbers of E6 frameshift mutations in the EqS#5-sgE6. In EqS#5-sgE5, EqS#5-sgLCR, and EqS#1-sgLCR, frameshift mutations were lower compared to wild-type sequences. Plotted are means and standard deviations from four biological replicates. Ctrl Fbs: BPV-1 negative control fibroblasts. EqS#1, 5: equine sarcoid-derived cells. LCR: long control region. nd: replacement of nucleotides without frameshift.

**Figure 3 viruses-15-01942-f003:**
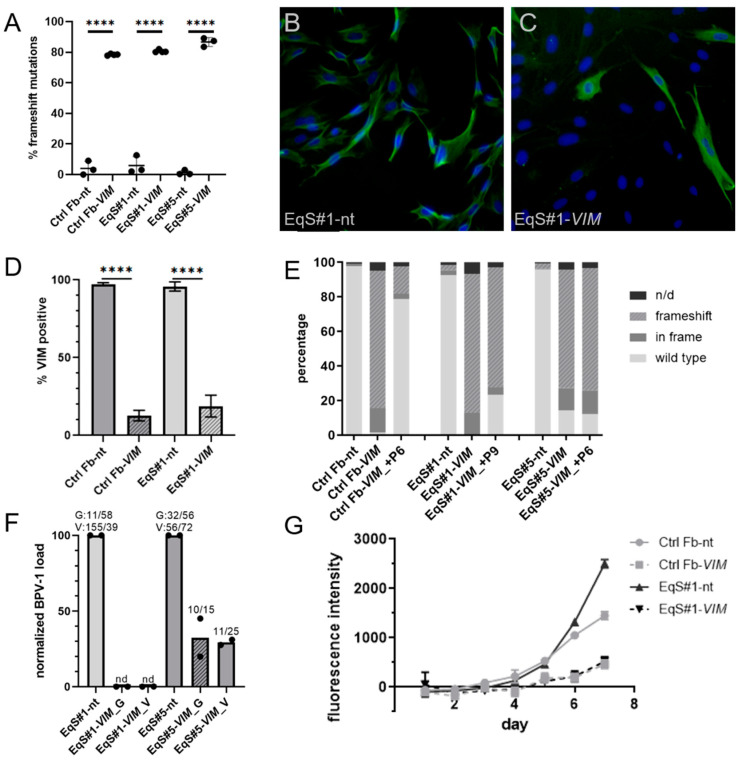
CRISPR/Cas9-mediated targeting of Vimentin in control and EqS fibroblasts. (**A**) TIDE analysis revealed a significantly (*p* < 0.0001) higher percentage of frameshift mutations in control and EqS fibroblasts transduced with VIM-sgRNA compared to NT-sgRNA-transduced cells. Plotted are means and standard deviations from three biological replicates. Analysis: ANOVA with Tukey’s multiple comparisons test. (**B**–**D**) Immunofluorescence confirmed a significantly lower percentage of VIM-positive cells in the fibroblasts transduced with VIM-sgRNA. Plotted are means and standard deviations from two biological replicates, each with three technical replicates. Analysis: ANOVA with Tukey’s multiple comparisons test. (**E**) A second TIDE analysis performed at a higher passage of VIM-sgRNA-transduced cultures revealed a selection in favor of VIM wild-type cells in ctrl Fbs. In EqS#1 and EqS#5 cells, a high percentage of VIM frameshift mutations persisted in higher passages. (**F**) Quantification of the BPV-1 load by two different qPCR methods (G = Ghent, E1; V = Vienna, E2) shows a pronounced reduction of the BPV-1 DNA in VIM-sgRNA-transduced EqS#1 and EqS#5 cells. Plotted are means and values (dots) from two biological replicates normalized to the viral load in NT-sgRNA-transduced cells. The numbers above the columns display the absolute BPV-1 load per cell for each replicate. (**G**) The proliferation assay revealed a reduced proliferation of VIM-sgRNA-transduced cells. Plotted are means and standard deviation from two biological replicates, each with three technical replicates. Ctrl Fbs: BPV-1 negative control fibroblasts. EqS#1, 4, 5, 6: equine sarcoid-derived fibroblasts. n/d: replacement of nucleotides without frameshift. nd: not detectable or lower than 0.2. **** *p* < 0.0001.

**Table 1 viruses-15-01942-t001:** Overview of sgRNAs.

sgRNA	Sequence 5′-3′	Target
NT ^1^	TGATTGGGGGTCGTTCGCCA	none targeting
*VIM* ^2^	GGTGGAGCGCGACAACCTGG	Vimentin exon 1
LCR	CCATCACCGTTTTTTCAAGC	BPV-1 long control region
E5	CATTTTGAGTGCTCCTGTAC	BPV-1 E5
E6	AGGTGTTCCAGTAACAGGTG	BPV-1 E6

^1^ The NT-sgRNA has neither homology to the horse nor to the BPV-1 genome. ^2^ sgRNA VIM targets the equine *VIM*.

## Data Availability

All research data of the study are available upon request to the corresponding authors.

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
