# Peer review of "CRISPR/Cas9-Mediated Targeting of BPV-1-Transformed Primary Equine Sarcoid Fibroblasts"

_viruses, 2023, doi:10.3390/v15091942_

Round 1

Reviewer 1 Report

This manuscript aims to provide proof of concept that CRISPR/CAS9 mediated oncogene or equine Vimentin gene (VIM) mutagenesis in primary tumor fibroblast cultures of BPV positive equine sarcoids (EqS) can be applied as treatment approach for these deadly and hard to treat equine tumors.

The given rationale for these studies is that CRISPR/Cas9 mediated oncogene knockout or reduction of episomal viral BPVgenomes lead to a reversion of PV oncogene driven carcinogenesis and or inhibition of cell viability/induction of cell death. Furthermore the equine VIM was explored as potential target to of CRISPR/Cas9 knockout to treat EqS.

Knockout approaches following Lentiviral delivery of the BPV E6 and E5 and LCR specific CRISPR/Cas9 machinery resulted reduction of BPV copy numbers. Knockout of VIM resulted eliminated episomal BPV genomes resulting in reduced proliferation or cultured EqS fibroblast that VIM targeting could be an effective treatment approach.    

While the rationale for the study is good, the results of the study emphasize a particular need for rigorous assay design and execution of the in vitro cell culture experimentation, as shown by the reduction of BPV load over time when culturing EqS fibroblasts restricting the time point to achieve relevant experimental results.

Nevertheless, some improvements need be made and some information to could be be added to strengthen the manuscript.

Specific questions and criticism:

1: The authors did not provide much information on the characteristics of their primary EqS fibroblast cultures regarding for purity of the cell pool or expression of specific markers associated with EqS. Cell characteristics in terms of BPV oncogene expression in the cultured cells and original tumors would be informative to assess the relevance of the experiments regarding the differences in the treatment response observed between individual EqS cultures.

2: As different responsiveness of the respective EqS fibroblast cultures to the BPV oncogene specific CRISPR/Cas9 treatment was observed in this study, the reason could lie could be related to small sequence variability in gRNA target sequences of individual BPV variant in different EqS samples. Therefore, I ask myself if the potential target sites within BPV E6 or E5 or LCR of the individual EqS samples might where analyzed to exclude that some point mutations already present before treatment limiting the efficacy of CRISPR/cas9 mediated mutagenesis. Such information might explain the differences in mutation frequencies presented in the manuscript. Therefore, pre-screening the BPV-DNA present in the initial tumor and cultivated cell for potential sequence variations in the respective target sites might be helpful to interpret the results as the autors stated: “BPV-1 episomes with partially deleted sequences and intratypic sequence variation were found in EqS tissue [11, 39, 40].”

3: The authors showed nicely that BPV oncogene and LCR as well as VIM targeting resulted in reduction or loss of BPV genomes. At the same time EqS fibroblast showed reduced growth kinetics when treated with anti-VIM-CRISPR system. Nevertheless treated  control fibroblasts where inhibited in their growth too, indicating that growth inhibition is not related to BPV genome content which seem to speak against such approach with regard to  possible side effects. 

4. Moreover, no growth kinetics of anti-BPV-oncogene or anti-LCR treated cells compared to mock treated cells have been shown making it difficult to the reader to estimate a benefit of such an approach to be used as future EqS treatment to eliminate tumor cells. As BPV copy number reduction seem to be efficient, the influence of BPV oncogene mutation induction and genome copy number reduction on cell vitality/ proliferation is an important information that is missing. Here more information on the effect on cell vitality and proliferation would be beneficial to the manuscript.

5. Maybe some discussion on how to translate the findings closer to a potential future application would be interesting for the reader.

6: in line 319 “in” appears twice in a row. Please revise the sentence.

Author Response

We thank the reviewer for the constructive and valid feedback on our manuscript and for recommending it for publication with minor revisions. In our revised manuscript, we have carefully addressed the reviewer’s points and included additional data in the manuscript (see attachments)

1: The authors did not provide much information on the characteristics of their primary EqS fibroblast cultures regarding for purity of the cell pool or expression of specific markers associated with EqS. Cell characteristics in terms of BPV oncogene expression in the cultured cells and original tumors would be informative to assess the relevance of the experiments regarding the differences in the treatment response observed between individual EqS cultures.

We thank the reviewer for this valid comment. For a comparative assessment of the tumors and the fibroblast cultures, PCR and rtPCR were conducted to assess DNA and mRNA of major BPV-1 oncogenes according to the protocols and references detailed in the revised methods section 2.4 (revised manuscript with changes marked: line 116-126) and supplementary table 2. The results are shown in supplementary figure 3-6 of the revised supplementary material. The reference to the supplementary figures was added in line 244-246 of the revised manuscript with changes marked.

We further assessed the purity of our cultures based on morphology. The corresponding data are referred to in line 239-240 of the revised manuscript. Supplementary figure 2 was added to the revised supplementary material and shows the morphology of Ctrl and EqS fibroblasts cultures. Furthermore, it should be noted that the Vimentin staining performed in our Vimentin targeting experiment revealed more than 95% Vimentin positivity of the fibroblasts of line EqS#1 and EqS#5 (Figure 3D), underlining the mesenchymal nature and purity of the cells contained in the culture.

2: As different responsiveness of the respective EqS fibroblast cultures to the BPV oncogene-specific CRISPR/Cas9 treatment was observed in this study, the reason could lie could be related to small sequence variability in gRNA target sequences of individual BPV variants in different EqS samples. Therefore, I ask myself if the potential target sites within BPV E6 or E5 or LCR of the individual EqS samples might where analyzed to exclude some point mutations already present before treatment limiting the efficacy of CRISPR/cas9 mediated mutagenesis. Such information might explain the differences in mutation frequencies presented in the manuscript. Therefore, pre-screening the BPV-DNA present in the initial tumor and cultivated cell for potential sequence variations in the respective target sites might be helpful to interpret the results as the authors stated: “BPV-1 episomes with partially deleted sequences and intratypic sequence variation were found in EqS tissue [11, 39, 40].”

We greatly appreciate the reviewer’s point. Before performing our experiments, we sequenced the gRNA target locus in all our tumors and carefully assessed the chromatograms. Moreover, the sequencing of the EqS fibroblasts transduced with non-targeting gRNA did not reveal differences in the target locus. The corresponding data were added to the revised supplementary material as supplementary figure 7-12. We refer to these data in the revised manuscript with changes marked in lines 268 to 271 and 286-288.

3: The authors showed nicely that BPV oncogene and LCR as well as VIM targeting resulted in reduction or loss of BPV genomes. At the same time EqS fibroblast showed reduced growth kinetics when treated with anti-VIM-CRISPR system. Nevertheless treated control fibroblasts where inhibited in their growth too, indicating that growth inhibition is not related to BPV genome content which seem to speak against such approach with regard to possible side effects.

The authors fully agree with the reviewer's point. The authors indeed do not consider VIM as a target for EqS treatment, and this is not stated in the manuscript. The authors rather used VIM to exemplify that a CRISPR-based targeting of host factors provides a valid approach to further study the relevance of potential host factors in equine sarcoid pathogenesis. To underline this, we adapted the discussion and added the sentence: “The antiproliferative effect on ctrl Fbs rather excludes VIM as a therapeutic target for EqS.” in line 387-388 of the revised manuscript with changes marked.

4: Moreover, no growth kinetics of anti-BPV-oncogene or anti-LCR treated cells compared to mock treated cells have been shown making it difficult to the reader to estimate a benefit of such an approach to be used as future EqS treatment to eliminate tumor cells. As BPV copy number reduction seem to be efficient, the influence of BPV oncogene mutation induction and genome copy number reduction on cell vitality/ proliferation is an important information that is missing. Here more information on the effect on cell vitality and proliferation would be beneficial to the manuscript.

The authors agree with this point and we integrated this information accordingly. The EqS Crispr/Cas9 BPV-1 E6, E5 and the LCR transduced cell lines displayed a stagnation in growth confining further expansion. Hence, performance of proliferation assays was not possible.

The sentence: “Other sgRNA transduced cells were viable but stagnated in growth thus confining further expansion” was added in lines 275 and 276 of the revised manuscript with changes marked.

5: Maybe some discussion on how to translate the findings closer to a potential future application would be interesting for the reader.

The authors appreciate the reviewer’s suggestion and refer to lines 399-402 of the revised discussion. “In this regards, our proof of concept study might be expanded to a CRISPR screening approach using gRNA libraries [51]. This enables to functionally test which host genes are essential for BPV infection and may thereby provide new mechanistic insights into the disease.”

6: in line 319 “in” appears twice in a row. Please revise the sentence.

The authors revised the sentence accordingly (line 325 of the revised manuscript with changes marked).

Reviewer 2 Report

The main question was informations about epigenetic players in cancer, it address a specific gap in the field, and new epigenetic information inside the model used. The study is completely and the conclusions are consistent with the evidence and arguments presented and they address the main question posed. The references are appropriate and I have no additional comments. It could be accepted in current form.

Author Response

The main question was information about epigenetic players in cancer, it addresses a specific gap in the field, and new epigenetic information inside the model used. The study is complete and the conclusions are consistent with the evidence and arguments presented and they address the main question posed. The references are appropriate and I have no additional comments. It could be accepted in current form.

We are grateful for the positive feedback provided and for considering our manuscript for publication. The authors agree that epigenetics are an important factor worthwhile investigating. This is underlined by the added data that indicate the absence of point mutations in the sgRNA target sequence (see revised supplementary material, supplementary figure 7-12).